# Why Should Pistachio Be a Regular Food in Our Diet?

**DOI:** 10.3390/nu14153207

**Published:** 2022-08-05

**Authors:** Raquel Mateos, María Desamparados Salvador, Giuseppe Fregapane, Luis Goya

**Affiliations:** 1Institute of Food Science, Technology and Nutrition (ICTAN-CSIC), Spanish National Research Council (CSIC), José Antonio Nováis 10, 28040 Madrid, Spain; 2Facultad de Ciencias y Tecnologías Químicas, Universidad de Castilla-La Mancha, Camilo José Cela n° 10, 13071 Ciudad Real, Spain

**Keywords:** pistachio, prevalent chronic diseases, nuts, nutritional value, health benefits, barriers, facilitator

## Abstract

The pistachio is regarded as a relevant source of biologically active components that, compared to other nuts, possess a healthier nutritional profile with low-fat content composed mainly of monounsaturated fatty acids, a high source of vegetable protein and dietary fibre, remarkable content of minerals, especially potassium, and an excellent source of vitamins, such as vitamins C and E. A rich composition in terms of phytochemicals, such as tocopherols, carotenoids, and, importantly, phenolic compounds, makes pistachio a powerful food to explore its involvement in the prevention of prevalent pathologies. Although pistachio has been less explored than other nuts (walnut, almonds, hazelnut, etc.), many studies provide evidence of its beneficial effects on CVD risk factors beyond the lipid-lowering effect. The present review gathers recent data regarding the most beneficial effects of pistachio on lipid and glucose homeostasis, endothelial function, oxidative stress, and inflammation that essentially convey a protective/preventive effect on the onset of pathological conditions, such as obesity, type 2 diabetes, CVD, and cancer. Likewise, the influence of pistachio consumption on gut microbiota is reviewed with promising results. However, population nut consumption does not meet current intake recommendations due to the extended belief that they are fattening products, their high cost, or teething problems, among the most critical barriers, which would be solved with more research and information.

## 1. Introduction

Nuts are regarded as a relevant source of biologically active components, mainly thanks to their great proportion of unsaturated and essential fatty acids as well as phenolic components [1,2,3]. In fact, in July 2003, the health claim: ‘‘scientific evidence suggests but does not prove that eating 1.5 oz (42.5 g) per day of most nuts, such as pistachios, as part of a diet low in saturated fat and cholesterol may reduce the risk of heart disease”, was ratified by the Food and Drug Administration of United States [4].

Since then, more studies have been carried out to determine the impact of frequent consumption of nuts on our health. Among them, the PREDIMED study [5] is one of the most valued interventions, which evaluated the effect of a Mediterranean diet supplemented with either 30 g of nuts or olive oil in 7447 participants at high risk of cardiovascular disease (CVD) in comparison to the control diet. Results showed that the Mediterranean diet that included the daily consumption of nuts reduced the risk of suffering a myocardial infarction, stroke, or death from a cardiovascular cause. Thus, improving glycaemic control, lipid levels, blood pressure, obesity biomarkers, and endothelial function, among others, could minimise the risk of suffering CVD [3,6]. Indeed, in a recent overview, the health benefits of nuts regarding the management of dysmetabolic conditions, such as obesity and type 2 diabetes mellitus (T2DM), closely associated with CVD, have been broadly proven [7].

Regarding the most consumed types of nuts, almonds and walnuts accounted for half of the total tree nut estimated consumption worldwide in 2019 (30% and 20%, respectively), followed by cashews, pistachios and hazelnuts, which accounted for 18%, 15% and 11%, respectively, according to International Nut & Dried Fruits [8]. Following Europe as the leading consumer, Asia and North America were the second and third largest consuming regions, with similar market shares. In the period 2010–2019, nuts were mainly consumed in high- and middle-income economies (56% and 39% of the world share for tree nuts in 2019), while peanuts were mostly consumed in middle-income countries, which represented 91% of the global share [8].

It is worth drawing attention to the pistachio among the nuts. *Pistacia vera* L. is a plant of the Anacardiaceae family which yields a distinctive green fruit of great gastronomic reputation called pistachio. Its cultivation began in western Asia but later spread to the Mediterranean basin through Iran [9]. Season 2020/2021 was largely an “on year” and, as a result, global production totalled over 1 million metric tons (in-shell basis), the highest amount of the last decade, representing a 54% increase from the previous year and a 68% increase over the previous 10-year average. The United States was the top supplier, accounting for 47% of the global share. It was the second “on year” in a row for Iran, and Turkey showed an exceptionally high bumper crop. The US, Turkey, and Iran produced 97% of the world’s pistachio. Syria (2%), Greece (1%), and others, such as Italy, Spain, Greece, Tunisia, Afghanistan, China, and Australia, account for the remaining 1% [8]. Although pistachio has been the least explored nut, there is broad evidence confirming its beneficial health effects and, particularly, its positive contribution to minimizing the risk for CVD [10,11], among other pathologies.

Due to the health benefits associated with the intake of nuts, their consumption is promoted as a regular food in most of the food-based dietary guidelines [12] in the context of a healthy diet. However, the population does not always consume them. For example, the Global Burden of Disease Study 2017 estimated 21 g per day as the optimal intake of nuts and seeds, while their actual consumption barely reached 12% of the recommended level [13].

This review aims to revalue pistachio consumption, the least explored nut but with the same or even more healthy potential than the rest due to its interesting nutritional composition. A review of the studies focused on evaluating the biological properties of pistachio, world consumption, and other uses will be developed. In addition, barriers and facilitators of pistachio consumption will be analysed to understand why the dietary requirements for nut consumption are not being met, attempting to shed light and help make it happen.

## 2. Nutritional Value of Pistachios

The pistachio is a low-water (3–6%) and nutritionally rich nut mainly because of its high fat (48–63%) and protein concentration (18–22%), together with the dietary fibre (8–12%; Table 1) [14,15]. In fact, the daily intake of nuts recommended (1.5 oz equivalent to 42.5 g; 4) in the form of pistachios is approximately 15% of the Dietary Reference Intake (DRI) for proteins, 11–18% of DRI for male and female respectively for dietary fibre, and 24% of DRI for fat. Lipids, although present in great amounts, have an equilibrated content of mono- (56–77%) and polyunsaturated (14–33%; Table 1) fatty acids, that may help to reduce LDL-cholesterol and hence the coronary heart disease risk [2,16,17].

Dry roasted pistachios have a lower fat content (45.82 g/100 g) than other nuts (Table 1), mainly monounsaturated fatty acid (24.53 g) followed by polyunsaturated fatty acid (13.35 g) and saturated fatty acid (5.64 g). Of the fatty acids, oleic is the main monounsaturated fatty acid (MUFA) in pistachios followed by linoleic acid (C18:2), which represents more than half of the total fatty acid content (<60 g) (Table 1). Pistachio shows a similar lipid profile to almonds and hazelnuts while walnut fat is headed by linoleic acid (54.0–65.0 g/100 g) followed by oleic (C18:2) and linolenic (C18:3) acids with similar contents. Pistachios are also a worthy source of vegetable protein (about 21%) as almonds (21%), and higher than other nuts, such as hazelnuts and/or walnuts. The amount of total carbohydrates is low to moderate (28%), but pistachios are rich in dietary fibre, mainly insoluble fibre (about 10% versus less than 1% of soluble fibre).

Pistachio also contains a remarkable content of minerals, such as magnesium, calcium, potassium, and phosphorus, and is documented as an important dietary source of potassium (1025 mg/100 g versus ~700 mg of almond and hazelnut or 450 mg of walnut per 100 g dry roasted nuts). Pistachios are rich in vitamins, especially vitamins C (5.60 mg/100 g) and E (mainly as γ-tocopherol) (2.17 g/100 g). Others, such as vitamin A, vitamin B (except B12), vitamin K, and folate, are existent in pistachios, which could contribute to their respective Recommended Dietary Allowance (RDA).

Furthermore, the analysis of phytochemicals in this kind of nut has shown the content of a diversity of bioactive polar and non-polar components, like tocopherols, phytosterols, and phenolics [3,15], reaching the top fifty foods possessing higher antioxidant activity [18]. Thus, pistachios are the nuts with the highest content of phytosterols compared with the other widely consumed nuts (2790 mg/Kg versus 1990–1130 mg/kg present in walnut, almond, and hazelnut), including β-sitosterol, Δ5-avenasterol, campesterol, and stigmasterol [19]. Pistachio is characterized by its carotenoid content, particularly luteolin and zeaxanthin with an amount of about 2760 μg/100 g and for β-carotene of about 200 μg/100 g. Compared with hazelnuts, pistachio exhibited 16-fold and eight-fold higher levels of lutein/zeaxanthin and β-carotene, respectively [20]. Finally, pistachios are also a rich source of phenolic compounds, which will be addressed in a separate section given their nutritional and biological relevance.

## 3. Antioxidant Phenolic Composition of Pistachios

Nutrient databases, like the Phenol-Explorer and the flavonoid and proanthocyanidin databases (USDA) [21,22] report the content of polyphenol of many foods. Furthermore, the concentration of various families of phenolic components in nuts, including pistachio, as well as their antioxidant capacity and evidence for healthy effects have been recently reviewed [1,23].

Walnuts, pecans, and pistachios are the kinds of nuts with the higher phenolic amounts, on the contrary almonds, peanuts, and hazelnut present lower levels (e.g., [1,23]. A variety-dependent content of total polar phenolics (TPP) is found in pistachios, ranging from 1600 mg/kg for the Kastel cultivars to more than three times more, 4900 mg/kg for Larnaka. As reported, polar phenolic components can be grouped into different families. Flavanols being the most abundant phenolics found (about 90% of total, from 1500 mg/kg to 4500 mg/kg), whereas the Kastel and Larnaka cultivars showed the lowest and highest concentrations, respectively [24]. Other families, e.g., anthocyanins (from 54 to 218 mg/kg), flavonols (from 76 to 130 mg/kg), flavanones (from 12 to 71 mg/kg), and gallotannins (from 4 to 46 mg/kg), are also measured. These findings are similar to those reported by Alasalvar & Bolling [1] or Chang et al. [23] (Table 2).

Nuts are rather consumed roasted because this operation enhance their color, flavor, and crunchy characteristics. However, roasting conditions may lead to a relevant decrease of the antioxidant capacity in certain nuts (walnut and hazelnut), but in pistachio and almond its activity apparently keeps stable or is modestly improved [23,25]. This effect may be justified by the decrease of phenolics due to effect of heating which is countervailed by the generation of antioxidant-active components formed by the Maillard reactions. Therefore, nuts must be roasted under appropriate settings to keep their great polyphenols content, antioxidant capacity, as well as sensory properties.

It is important to highlight that the antioxidant capacity shown by different nuts depends on the type of assay employed. Therefore, it is recommended to use more than one type of antioxidant assay to take into account the different antioxidant mechanism and the restrains of each method [26]. In fact, various in vitro chemical assays (e.g., ABTS, FRAP, DPPH, ORAC, etc.) and biological methods (generally based on the oxidation of lipoprotein) are employed to measure the antioxidant capacity of nuts [23].

Regarding the TPP antioxidant activity of pistachios, a wide cultivar-dependent interval was obtained; showing Larnaka the greatest Trolox Equivalent Antioxidant Capacity (TEAC) for both ORAC (330 mmol/kg) and DPPH (35 mmol/kg) methods. On the contrary, Kastel and Kerman are the cultivars with the lowest TEAC data, 89 and 61 mmol/kg (ORAC), and 13 and 10 mmol/kg (DPPH) respectively [24]. These findings are in agreement with those observed by evaluating the DPPH antioxidant capacity of the Uzun Turkish pistachio cultivar [27,28]. Furthermore, Wu and Prior [29] created a database concerning ORAC data in foods, giving 76 mmol/kg for pistachio nuts, similar to the value obtained for Kastel and Kerman varieties (Table 2).

## 4. Health Benefits of Pistachio Consumption

Pistachios have an exciting nutritional profile compared with the rest of the nuts due to their lower energy content and highest levels of γ-tocopherol, phytoesterols, carotenoids, minerals, such magnesium and potassium, and vitamins K and B. The nutrients mentioned above undoubtedly contribute to the evidence that the regular intake of pistachios improves health [15]. Cardiometabolic disease involves dyslipidemia, insulin resistance, hypertension, and excessive visceral fat, which are behind T2DM and CVD [30]. Epidemiological and/or clinical trials have shown that nut consumption has a positive influence on health, by reducing the risk of suffering CVD [15,31,32], hypertension [33], T2DM [34] and obesity [35], among others. These beneficial effects are a consequence of their unique composition since nuts are nutrient-dense foods with healthy MUFA and PUFA fatty acid composition, dietary fibre, high-quality vegetable protein, vitamins, and minerals, along with carotenoids, phytosterols, and phenolic compounds previously described, with recognized benefits to human health [3,10,11,36]. Regarding pistachio, there is wide evidence confirming its beneficial health effects and, particularly, its positive contribution to minimizing the risk for CVD [10,11].

### 4.1. Effects of Pistachio Consumption on Blood Lipids

Hyperlipemia is an established risk for CVD. Pistachio intake has been related with the improvement of lipid profile, decreasing total cholesterol (TC) concentration [37,38,39,40], TC/high-density lipoprotein (HDL) ratio and low-density lipoprotein (LDL)/HDL ratio [37,38,39,41] in the pistachio-supplemented cluster compared with the control one, in both healthy [38,39,40] and patients with moderate hypercholesterolemia [37,41]. LDL concentrations also decreased significantly in the pistachio-supplemented group in some studies [39,40,42], whereas others observed a non-significant reduction [37,38,43]. Additionally, Sheridan et al. [41] observed a significant increase in circulating HDL concentration in those subjects who consumed pistachios. Recently, a systematic review of epidemiological evidence developed by Lippi et al. [44] showed the beneficial effects of pistachio intake for improving the blood lipid profile (Figure 1).

Clinical trials were developed to study the effect of pistachio consumption on the concentration and size of lipoprotein subclasses (small, medium and large), other blood lipid markers of atherosclerosis different from the classical lipid profile [43,45], found a significant antiatherogenic variation of lipoprotein subclasses. Thus, much evidence suggests that pistachios may improve the blood lipid profile, contributing to decreased cardiovascular risk. Most interestingly, in a network meta-analysis of clinical trials comparing the effects of different types of tree nut consumption on blood lipids published this year, Liu et al. [46] concluded that diets enriched with pistachio and walnut could be better alternatives for lowering tryglycerides (TGs), LDL cholesterol, and TC compared to other nut-enriched diets. Another conclusion from this study is that more clinical trials are needed to confirm those claims and suggest changes in dietary and nutritional patterns for the ordinary population. Last year, a systematic review and meta-analysis of randomized controlled trials strongly proposed that pistachios may improve lipid profiles (TC, LDL, TG) and protect against cardiometabolic diseases [47]. Early this year, a comprehensive meta-analysis of randomized controlled trials (RCT) studying the effects of nut consumption on blood lipid profile concluded that, although there was no overall effect of nut consumption on lipid profile, pistachio consumption may reduce TC levels [48] (Figure 1).

### 4.2. Effects of Pistachio on Blood Pressure and Endothelial Function

Several prospective studies have revealed an inverse relationship between nut consumption and blood pressure (BP) or hypertension. In the particular case of pistachio, a beneficial effect on BP has also been observed in a clinical trial conducted on 28 dyslipidaemic individuals who followed for four weeks either a low-fat control diet, a diet having 10% of the total energy from pistachios, or a diet with 20% of the total energy from pistachios. A significant systolic blood pressure (SBP) reduction was observed, particularly after following the diet supplemented with 10% of the total energy as pistachios, thus, no dose-dependent effect was observed. In addition, no difference in diastolic blood pressure (DBP) was recorded [49]. Another more recent study conducted on T2DM subjects by Sauder et al. [50] showed a decrease in SBP after consuming a diet with 20% energy from pistachios for four weeks. However, three additional controlled feeding trials evaluated the BP lowering effects of pistachios as a secondary outcome and non-significant differences in both SBP and DBP between those subjects supplemented or un-supplemented with pistachios were observed [40,41,51]. Finally, a recent review and meta-analysis of more than 20 RCTs found that despite the intake of mixed nuts may reduce DBP, pistachios seemed to have the most potent effect on reducing both DBP and SBP [33]. Therefore, although there is some evidence that suggests that pistachios may reduce BP the non-consistent results encourage continuing to investigate this aspect (Figure 1).

Nuts consumption also improves the endothelial function. The antiatherogenic effect of hazelnut before and after consumption in 21 hypercholesterolemic subjects was investigated. The consumption of a hazelnut-enriched diet (contributing 18–20% of the total daily energy intake) for four weeks significantly improved flow-mediated dilation (FMD), TC, TG, LDL, and HDL as well oxidized-LDL (oxLDL), C-reactive protein (CRP), and soluble vascular cell adhesion molecule-1 (sVCAM-1) compared with the control diet, confirming the antiatherogenic effect of hazelnut-enriched diets by improving endothelial function, preventing LDL oxidation and inflammatory markers, in addition to their lipid and lipoprotein-lowering effects [52]. Another study conducted by Katz et al. [53] evaluated the effects of daily walnut consumption on endothelial function and other cardiac risk biomarkers in 46 overweight adults with visceral obesity. Results revealed that a daily intake of 56 g of walnuts for eight weeks improved endothelial function, without weight gain, by improving FMD compared with the control diet, as well as beneficial trends in SBP and maintenance of anthropometric values. Recently, a review summarized the effects of tree nut and peanut intake on vascular function, excluding FMD. A total of 16 studies were evaluated, although very heterogeneous in terms of dose, length of supplementation, study designs, etc. Conclusions were discrepant. Ten studies provided no significant changes, and six studies (one acute and five chronic studies) informed improvements in at least one measure of vascular function. In summary, nuts have the potential to improve vascular function, but the development of future studies is necessary to expand existing knowledge [54] (Figure 1).

Regarding pistachio, Kasliwal et al. [55] evaluated its effect on vascular health in 60 adults with mild dyslipidemia after consuming 80 g (in-shell) pistachios for 12 weeks. Results showed that usual intake of pistachios not only improved glycaemic and lipid parameters but also produced improvements in vascular stiffness and endothelial function (carotid-femoral and brachial-ankle pulse wave velocity). In a recent meta-analysis of randomized, controlled-feeding clinical studies, Fogacci et al. [56] reviewed the effect of pistachio on brachial artery diameter and flow-mediated dilatation and suggested a significant effect of pistachios on endothelial reactivity, affecting brachial artery diameter but not flow-mediated dilatation. Recently, a comprehensive review [57] and several systematic reviews and meta-analyses Ghanavati et al. [58] demonstrated that pistachio consumption can elicit a beneficial effect on some cardiometabolic risk factors, and Asbaghi et al. [59] suggest the efficacy of pistachio consumption to reduce SBP. Finally, early this year, in the latest review dealing with the effect of dietary polyphenols on vascular health, Grosso and coworkers concluded that hypertension is counteracted by a plant-based dietary pattern, including pistachio, for example, since no single food is enough to control hypertension [60] (Figure 1).

### 4.3. Effects of Pistachio on Glucose Metabolism

In recent years, animal studies have unequivocally shown the positive effect of pistachio feed on glucose homeostasis in altered situations such as diabetes and metabolic syndrome (MetS). Thus, the consumption of pistachio has favorable effects on avoiding hypertriglyceridemia, hyperglycemia, hypercholesterolemia, and inflammation characteristic of a MetS induced by a fructose overload [61].

Recently, the pistachio extract administered to diabetic rats for three weeks significantly improved the lipid profile, oxidative stress, and inflammation process by reducing lipid peroxidation and increasing total antioxidant capacity [62]. In the same year, a pistachio hull hydro-alcoholic extract, along with aerobic exercise, improved passive avoidance memory in streptozotocin-induced diabetic rats [63]. More recently, a pistachio extract also reverted most parameters altered by streptozotocin-induced diabetes in rats; not only markers related to glucose homeostasis, but also those associated to ovary damage and oxidative stress [64].

Several epidemiological studies and clinical trials suggested that the regularity of nut consumption is inversely related to an increased risk of T2DM [65,66,67,68,69,70], mainly attributed to the relatively high content in dietary fibre, the presence of healthy fats and antioxidants components [69]. Specifically, the effect of consuming pistachio alone or combined with meals was evaluated on postprandial glycaemia [71,72]. Pistachios consumed alone had a minimal effect on postprandial glycaemia, and when 28, 56, or 84 g of pistachios were taken with a carbohydrate meal attenuated in a dose-dependent manner the glycaemic response [71]. These authors evaluated pistachio consumption’s acute effects on postprandial glucose and insulin levels in a randomized crossover study conducted on subjects with MetS. Results showed that compared with white bread, pistachio with bread decreased postprandial glycaemia levels and increased glucagon-like peptide levels [72]. Feng et al. [73] confirmed the positive effect that 42 g of pistachio had on postprandial glycemic and gut hormone responses in women with gestational diabetes mellitus or gestational impaired glucose tolerance, by inducing significantly lower postprandial glucose, insulin, and gastric inhibitory polypeptide (GIP) but higher glucagon-like peptide-1 (GLP-1) levels compared to 100 g of whole-wheat bread.

In a controlled clinical trial, healthy young men were randomly addressed to a Mediterranean diet or a Mediterranean diet where monounsaturated fat content was replaced by pistachios (equivalent to 20% of daily caloric intake) for four weeks. Subjects who followed the Mediterranean diet supplemented with pistachios showed a significant decrease in fasting plasma glucose concentrations as compared to the control [40]. Following the Mediterranean diet-based nutritional intervention, a study carried out by Melero et al. [74] confirmed the positive influence on the pregnancy of this dietary pattern by reducing the rate of gestational diabetes mellitus in 600 Spanish women who had to intake about 25–30 g of pistachios at least 3 days a week. Besides, a nutritional clinical trial based on the Mediterranean diet during pregnancy seems to be related with a decrease in the offspring’s hospital admission, especially in women with pre-gestational body mass index (BMI) < 25 kg/m2 [75]. Later, the consumption of 0, 42, and 70 g/day of pistachios by 70 subjects with MetS for 12 weeks, evidenced a downward trend in blood glucose levels in subjects who adhered to pistachio interventions compared to the control diet, but without reaching significant differences [51]. A similar study carried out with subjects with MetS randomized to either an unsalted pistachios diet (20% dietary energy) or a control diet for 24 weeks showed a significant decrease in glucose levels but not blood insulin levels [76]. While Hernandez-Alonso et al. [43] demonstrated that pistachios have glucose- and insulin-lowering effect in a RCT with 54 pre-diabetic subjects who consumed for four months a pistachio-supplemented diet in comparison with a control diet. The effect of replacing carbohydrate consumption with mixed nut consumption (75 g/d) that included pistachios in 117 T2DM subjects for three months showed a significant decrease in glycated hemoglobin (HbA1c) levels, improving glycaemic control [77]. Results were similar in a crossover trial with 48 diabetic participants after three months of pistachio consumption [78]. Interestingly, chronic pistachio consumption reduced oxidative damage to DNA and increased the gene expression of some telomere-associated genes, reversing certain deleterious metabolic consequences of prediabetes [79]. More recently, pistachio, among other Mediterranean products, has been recommended a promising source of multi-target agents in the treatment of MetS [80]. Despite the encouraging results reported for glucose metabolism in fasting conditions or postprandial status, additional studies are necessary to test pistachio consumption’s long-term effects on insulin resistance and T2DM prevention and/or control. In a recent systematic review and meta-analysis of pistachio on glycemic control and insulin sensitivity in patients with T2DM, prediabetes and MetS, the authors concluded that pistachio nuts might cause a significant reduction in fasting blood glucose and HOMA-IR (homeostatic model assessment for insulin resistance), although HbA1c and fasting plasma insulin might not significantly improve in patients suffering from or at risk of T2DM [81]. Finally, a RCT published early this year has reported that a Mediterranean diet with additional extra virgin olive oil and pistachios decreases the occurrence of gestational diabetes mellitus [82] (Figure 1).

### 4.4. Effects of Pistachios on Satiety Regulation and Body-Weight Control

Although nuts, including pistachios, are still perceived by the general public to be fattening because of their high-fat content, there are several epidemiological studies that have provided strong evidence that nut consumption is associated with neither weight gain nor an increased risk of obesity [83,84]. Regarding pistachio, Li et al. [85] evaluated the effects of pistachio snack consumption on body weight and lipid levels in obese participants, which were randomly assigned to consume isocaloric weight reduction diets for 12 weeks with an afternoon snack of either 53 g of salted pistachios or 56 g of salted pretzels. Both groups lost weight, but the pistachio-supplemented group showed a higher BMI reduction than the pretzel-supplemented group. Regarding the already mentioned study of Wang et al., [51], results showed that the supplementation with pistachios (42 or 70 g/d) for 12-weeks did not lead to weight gain or an increase in waist-to-hip-ratio in Chinese subjects with MetS. Likewise, Gulati et al. [76] evaluated the effects of pistachios on body composition in 60 individuals with MetS randomised to either the pistachio (20% dietary energy) or control group for 24 weeks, and no significant differences were detected in body weight, although a significant decrease in waist circumference was observed. However, Parham et al. [78] found a significant reduction in BMI after 50 g pistachio consumption for 12 weeks compared to the control diet.

Other clinical studies where apart from studying the effect of pistachio consumption on biomarkers of cardiovascular health, also evaluated the impact on weight or/and BMI, did not observe differences between participants belonging to the supplemented diet with nuts and the control group [38,39,40,41]. A more recent study developed by Fantino et al. [86], concluded that the daily intake of 44 g pistachios improved quality diet without disturbing body weight in healthy women. While Rock et al. [87] evaluated the impact of consuming 42 g/d of pistachios by non-diabetic overweight/obese adults assigned to a four-month behavioural weight loss intervention against a similar group without consuming pistachios, observing a similar reduction of weight, BMI, and waist circumference (Figure 1).

Among the various explanations of why the consumption of pistachios does not induce overweight, being a very energetic food, is their high satiating power, inefficiency in the absorption of the energy they contain, a possible increment in energy expenditure at rest and an increase in fat oxidation (revised by Tan et al. [84]). Moreover, the crunchy physical structure of nuts in general, and pistachio in particular, has demonstrated its positive influence on satiety [88]. In this sense, recently, a randomized controlled pilot study was carried out to assess the effects of a daily pistachio afternoon snack on next-meal energy intake, satiety and anthropometry in 30 healthy French women were tutored to consume either 56 g of pistachios or 56 g of isoenergetic/equiprotein savoury biscuit as an afternoon snack. Results revealed that both afternoon snacks provided a similar subjective feeling of satiety, and pistachios consumption did not affect body weight or composition [89]. Lately, pistachio treatment in obese mice fed a high-fat diet showed neuroprotective effects, including decreased brain apoptosis, decreased brain lipid, and oxidative stress with the improvement of mitochondrial function [90]. A recent meta-analysis of RCTs by Xia et al. [91] reported that a diet with pistachios reduced BMI and had no significant effects on body weight and waist circumference. Indeed, recent research has focused on pistachio’s applications as a plant-based snack, particularly for appetite control and healthy weight management [92]. Due to the importance of satiety regulation on body-weight control, clinical trials with pistachios need to be carried out in the future to establish this aspect fully.

### 4.5. Effect of Pistachios on Inflammatory State

Chronic low-grade inflammation has been related with insulin resistance, diabetes, atherosclerosis, obesity and MetS. A few research studies have evaluated the effects of nut intake on inflammation with different results. A previous study suggested that a diet supplemented with pistachio improved some inflammation biomarkers in healthy young men; by decreasing serum interleukin 6 but without changing CRP and tumour necrosis factor-alpha (TNFα) levels [40]. Recently, proanthocyanidins extracted from Sicilian pistachio was the major bioactive able to modulate the inflammatory response of human intestinal epithelial cells through the inhibition of nuclear factor kappa B (NF-κB) activation [93]. More recently, the anti-inflammatory effect in vivo of pistachio was shown in a rat model of ulcerative colitis inflammation [94]. In the same year, the results of another study showed that usual pistachio intake improved inflammation in obese mice, probably due to the positive modulation of the microbiota composition [95]. More recently, an aqueous leaf extract obtained from *Pistacia lentiscus* improved acute acetic acid-induced colitis in rats, which was associated with its ability to reduce inflammation and oxidative stress [96] (Figure 1).

### 4.6. Effects of Pistachios on Oxidative Stress

Pistachios contain polyphenols that act as radical scavengers, neutralising reactive oxygen species (ROS) and enhancing endogenous antioxidant defences. Many studies have demonstrated the antioxidant activity of extracts obtained from different pistachio parts in both in vitro [97,98,99,100,101] and in vivo models, such as in animals [64,97,99,102,103,104,105,106,107,108] and humans [7,15,54,55,76,109]. Therefore, antioxidants present in pistachios could have significant effects on the regulation of oxidative stress and a reduced risk of chronic diseases. Remarkably, previous research on the antioxidant capacity and chemo-preventive potential of pistachio phenolic compounds in cell culture models has been performed in cell types from thyroid, lung, skin, and monocyte/macrophage, but not in endothelial cells, which is the more suitable cell type to study endothelial dysfunction and its potential effect on cardiovascular capacity. This inattention should be repaired in the near future.

Regarding human studies, a study carried out on 44 healthy volunteers showed an increased blood antioxidant potential determined by the evaluation of biomarkers of lipid peroxidation in those participants consuming pistachios compared with those assigned to a control diet without nuts [38]. A cross-over RCT developed by Kay et al. [110] on 28 hypercholesterolaemic adults showed that the consumption of diets containing 10% and 20% of energy from pistachios increased γ-tocopherol, lutein and β-carotene concentrations in serum, and decreased oxLDL concentration compared with the un-supplemented group. Sari et al. [40] evaluated the effect of the Mediterranean diet supplemented with pistachios by replacing the monounsaturated fat content on thirty-two healthy young men for four weeks in a prospective study. Results revealed an increase in total antioxidant status and superoxide dismutase, along with a decrease in inflammation and other oxidative markers. In a more recent randomized, double-blind, placebo-controlled trial, the antioxidant efficacy of a *Pistacia lentiscus* supplement in inflammatory bowel disease was assessed [111]. 60 patients were randomly allocated to *Pistacia lentiscus* supplement (2.8 g/day) or to placebo for three months and oxLDL, oxLDL/HDL, and oxLDL/LDL decreased significantly in the intervention group, confirming its antioxidant activity. These afore-mentioned results have demonstrated the beneficial effects of pistachios on the factor risks of CVD (Figure 1).

### 4.7. Effects of Pistachios on Cancer

In cell culture studies, mastic gum resin has shown anticancer effect in bile duct cancer (cholangiocarcinoma) (KMBC), pancreatic carcinoma (PANC-1), gastric adenocarcinoma (CRL-1739), and colonic adenocarcinoma (COLO205) cells [112]. *P. lentiscus* extract showed moderate activity against liver cancer [113] as well as breast cancer [114,115]. Raw and roasted pistachio showed a chemo-preventive potential regarding colon cancer [116]. The essential oil from *Pistacia lentiscus* aerial parts induced oxidative stress and apoptosis in human thyroid carcinoma cells [98]. Pistachio green hull extract induced apoptosis through multiple signaling pathways by causing oxidative stress on colon cancer cells [117]; whereas the cytotoxic fraction from *Pistacia vera* red hull ethyl acetate extract showed anticancer activity on breast cancer both in cell culture and in mice [118]. Mastiha is a natural aromatic resin obtained from the trunk and branches of the mastic tree (*Pistacia lentiscus* L. var latifolius Coss or *Pistacia lentiscus* var. Chia). Similar to the results reported with green hull extracts by Koyuncu and coworkers, mastic has shown a powerful anticancer effect in colon cancer cells [119]. In the same year, mastic gum essential oils exhibited selective cytotoxicity against SK-MEL-30 (melanoma), A559 (lung), PANC-1 (pancreatic), and U87MG (glioblastoma-astrocytoma) human cells lines, whereas HeLa (cervix adenocarcinoma) cells exhibited more sensitivity to the treatment [120]. In a very recent overview of pistachio effects on human health [121], beneficial effects on inflammation and oxidative stress were generally reported for mastic consumption, while a study developed in a cancer bioassay model showed an increase in biomarkers associated with the formation of hepatic preneoplastic lesions after mastic administration [122]. In contrast, mastic exhibited an antihepatotoxic activity in carbon tetrachloride-intoxicated rats, by reducing alkaline phosphatase activity and bilirubin levels [123]. Moreover, recently, a current systematic review and meta-analysis of observational studies showed the inverse association between pistachio consumption and the cancer risk and mortality [124]. Finally, early this year, the nanoliposomal formulation of pistachio hull extract showed promising anti-cancer effects through Bax/Bcl2 modulation [125] (Figure 1).

### 4.8. Effect of Pistachios on Intestinal Microbiota

Dietary fibre and phytochemicals present in nuts can reach the proximal colon and modulate the microbiota composition. Effects of almond and pistachio consumption on gut microbiota composition were evaluated in a crossover RCT for 18 days with 0, 1.5, and three servings/day. Pistachio showed a higher effect than almond on gut microbiota, increasing the number of potentially beneficial butyrate-producing bacteria, whereas *bifidobacteria* were unaffected [126]. A more recent study led by Yanni et al. [127] in diabetic rats demonstrated that dietary pistachio for four weeks restored normal flora and enhanced the presence of beneficial microbes (*lactobacilli* and *bifidobacteria*, among others) in the rat model of streptozotocin-induced diabetes. Likewise, chronic intake of pistachio for 16 weeks significantly enhanced the level of healthy bacteria genera such as *Parabacteroides*, *Dorea*, *Allobaculum*, *Turicibacter*, *Lactobacillus*, and *Anaeroplasma*, and decreased bacteria related to inflammation, such as *Oscillospira*, *Desulfovibrio*, *Coprobacillus*, and *Bilophila* in mice fed a high-fat diet [95]. Finally, Creedon et al. [128] carried out a systematic review of RCT to check the impact on gut microbiota and gut function that nut consumption has in healthy people. Seven RCTs were eligible, and only one of them was developed with pistachios. Nut consumption significantly increased *Clostridium*, *Dialister*, *Lachnospira* and *Roseburia* and significantly decreased parabacteroides. Nonetheless, it is convenient to carry out new studies to expand knowledge about the effect of pistachio on intestinal microbiota and gain robust conclusions (Figure 1).

## 5. Consumption and Uses of Pistachios

Regarding the estimated world consumption of pistachios (in-shell basis) per capita (Kg/year), the top five countries were Turkey (2.06 kg/year), Syria (1.83 kg/year), Israel (1.79 kg/year), Spain (1.43 kg/year), and Germany (0.98 kg/year) in 2019 [8].

Nuts in general, and pistachio in particular, are consumed fresh, roasted, or as salty snacks. Furthermore, gourmet and functional oils, flours, plant-based milk substitutes, spreads, cereal bars, and bakery goods, among others, may be made with nuts, with high requirements in terms of nut quality. Thus, nut oils are obtained by mechanical pressing without refining, and due to their sensorial attributes, the pistachio oil is an appreciated product from the gastronomy point of view (as a substitute for butter or margarine, as a dressing for vegetables, and as an ingredient of haute cuisine) [129]. The composition and properties of virgin pistachio oils processed employing different cultivars were characterized by Ojeda-Amador et al. [130] in 2018; highlighting the attractive added value of this type of vegetable oil. Recently, the development of functional edible oils enriched with pistachio and walnut phenolic extracts has been described [131], which constitute an exciting strategy to increase the phenolic intake in the diet. A phenolic concentration of 340–570 mg/kg was reached in the different enriched edible oils, showing a great antioxidant activity (up to 54 mmol/kg Trolox, as DPPH). Furthermore, the functional properties and stability of these formulations have been improved employing emulsion and micro emulsion systems [132]. Nut flour has aroused interest as a substitute for wheat flour for gluten-free formulations to use in bakery products such as bread, cakes and cookies [133] and with an interesting nutritional profile. Sanchiz et al. [134] evaluated pistachio flour after combining heat and pressure processing, resulting in a higher phenolic content and excellent physical properties of pistachio flour. Non-dairy milk alternatives answer the needs of people with milk protein allergy and lactose intolerance, and its market is growing at breakneck speed. The pistachio represents an excellent raw material for obtaining vegetable-based milk, as described by Sethi et al. [135]. Nut spread has also been formulated with pistachio, which constitutes an excellent alternative to increase its added value [136]. In the same line, pistachio has been used as a predominant ingredient in the formulation of cereal bars, which is very well accepted by consumers [137].

Furthermore, pistachio oil is a valued ingredient for cosmetic care formulations, e.g., shampoos, hair conditioners, soaps, skin creams, and lip balms, [129]. Recently, distilled leaves from pistachio have been reported as a potential cosmeceutical ingredient because of their transdermal diffusion and anti-elastase and anti-tyrosinase activities [138].

Such commercial presentations enlarge the use of pistachio as value-added ingredients, motivating their cultivation and production. Pistachio constitutes an ideal ingredient for new functional foods and satisfies the needs of more limiting diets, such as vegetarianism or veganism [139].

## 6. Barriers and Facilitators to Pistachio Consumption

Population nut consumption does not meet the current recommendation of intake despite their widespread demonstrated health benefits. In this sense, a recent review identified some barriers and facilitators to nut consumption [140], which should be taken into account in elaborating future strategies to encourage nut intake.

The most extended feeling about nut consumption is the belief that eating nuts would cause weight gain, despite the cumulative results reporting no effect on it [51,76,78,85]. An interesting and recent study that evidence this issue was that developed by Brown et al. [141], who evaluated the nut perceptions among the general public. Results revealed that over half the respondents reported they would eat more nuts if they were advised to do it by a dietitian or doctor, and the most frequently selected deterrent to increasing nut consumption was the potential weight gain (66%). Likewise, a survey of 710 interviewees in New Zealand identified as a barrier to regular nut consumption the perception that nuts are high in fat, which could affect negatively on control weight [142]. Another common barrier to nut consumption was their high cost. The study developed by Brown et al. [141] in 710 people found the cost (67%) as the most frequently selected deterrent to increase nut intake. Recently, a survey of 124 participants from the United States [143] highlighted an inverse relationship between the price and the intention to consume nuts. Dentition issues may act as a possible barrier to regular nut consumption, as was, for example, reported in the New Zealand study where it was mentioned as a common barrier to nut intake [142]. Other presentations of nuts, such as nut butter, would constitute interesting alternative forms of nuts for those with poor dentition. Finally, allergy to nuts constitutes an important nut intake barrier, as was reported by Australian health professionals in the study developed by Tran et al. [144]. In addition, this barrier is extrapolated to people who lived with or were in close contact with someone allergic to nuts, since up to 8% of participants compatible with this situation declared not to consume nuts or nut butter.

Conversely, the healthy properties associated with nut consumption constitute a strong facilitator, but certain demographic characteristics such as a higher level of education or socio-economic status [145,146,147], healthier lifestyle [145,148], and higher levels of physical activity [145], are determining a higher intake of nut. Likewise, increased age appears to be related to higher nut consumption [145,148]. In contrast, intake of nut butter may be higher among younger people [147]. Health professionals have a key role in encouraging nut intake, although their knowledge about health benefits derived from nut consumption could be improved.

The aforementioned data could help to identify the most vulnerable groups to not meeting the recommendations for the nut intake, very important to design targeted dietetic strategies to reach the minimal pistachio consumption.

From the evidence given and discussed in this review, it is clear why pistachio should be a regular food in our diet, since there are several well-documented health benefits related to the consumption of this kind of nut. However, this is probably clear to food and nutrition experts, just yet to reach the general population.

To increase pistachio consumption several strategies should be used. To effectively inform the population about the health benefits of its consumption, specific promotion campaigns should eb designed focused on different target consumers, e.g., kids, teenagers, adults, sportsmen/sportswomen, elderly, and so on. As mentioned, dietitians or doctors should be involved to increase people’s confidence in this message.

Moreover, campaigns should also promote the consumption of the different types of pistachio’s products, several of them described above. The fusion between gastronomy and nutrition is generally a successful story; increasing the number of recipes using pistachios as an ingredient but also developing new ways of employing them, such as virgin oils, nut butter, bakery products, etc.

Well-designed educational tasks, such as MyPlate or similar food guidelines, should reinforce nut consumption, that of pistachio in particular in this case, within the fruits and vegetables food group.

Since nuts are consumed roasted because it enhances their sensory characteristics, it should also be very relevant to advance the optimization of roasting conditions with the aim of maximizing the effects of pistachio’s bioactive polar and non-polar components; which must also involve a proper consumer’s sensory evaluation to ensure that preferences for the new product remain still high.

Finally, as commented in this review, a complementary strategy to ensure an adequate intake of pistachio’s bioactive components should involve the development of functional foods that contain phenolic extracts.

## 7. Conclusions

The pistachio is valued as an important source of bioactive components that, compared to other nuts, possesses a healthier nutritional profile with low-fat content composed mainly of MUFA, a high source of vegetable protein and dietary fibre, remarkable content of minerals, especially potassium, and an excellent source of vitamins, such as vitamins C and E. Its phytochemicals rich composition such as tocopherols, carotenoids and, importantly, phenolic compounds make pistachio an interesting food for its potential to prevent prevalent pathologies. The first health claim approved by the United States FDA in 2003 informed that there is scientific evidence that suggests, although does not demonstrate, that eating 1.5 oz (42.5 g) each day of nuts within a low saturated fat and cholesterol diet, may diminish the heart disease risk, was a turning point for nuts. A growing interest in confirming nuts effect on health has led to an increase in the quantity of research on nuts. Although pistachio has been less explored than other nuts (walnut, almonds, hazelnut, etc.), some studies provide evidence of its beneficial effects on CVD risk factors beyond the lipid-lowering effect. Different studies showed that diets supplemented with pistachios present a preventive effect to T2DM due to its effect of improving markers of glucose homeostasis, decreasing oxidative stress, alleviating post-prandial hyperglycemia and reducing the rate of gestational diabetes, among others. Regarding the most concern issue regarding pistachio intake, recent results showed neither weight gain nor obesity risk, but rather the opposite. A tendency to decrease and/or maintain weight as well as improve and/or maintain anthropometric parameters was observed and, in part, due to its positive effect on controlling appetite and enhancing satiety feeling, although more studies are needed to evaluate this aspect thoroughly. Likewise, the pistachio could play an important role in cancer prevention due to its ability to induce cytotoxicity and apoptosis in neoplastic cells, as well as the modulatory activity of signaling pathways involved in the regulation of cancer observed in carcinogenic cell models. The phytochemicals would be behind the demonstrated anti-inflammatory and antioxidant activities associated with pistachios, which would explain the ability to prevent the chronic diseases mentioned (CVD, T2DM, obesity, and cancer). Finally, there are more and more studies that inform about the pistachio’s ability to modulate the activity of gut microbiota, increasing the abundance of beneficial bacteria, although more studies are needed to gain robust conclusions.

This review provides enough information and evidence of the usual pistachio intake’s health benefits. However, specific barriers continue to make it difficult to reach the intake requirements for nuts, such as the belief that it is a fatty product, the cost, or teething problems. Except for the allergenic problems, the rest are solvable with more research and information from the population and health professionals.

## Figures and Tables

**Figure 1 nutrients-14-03207-f001:**
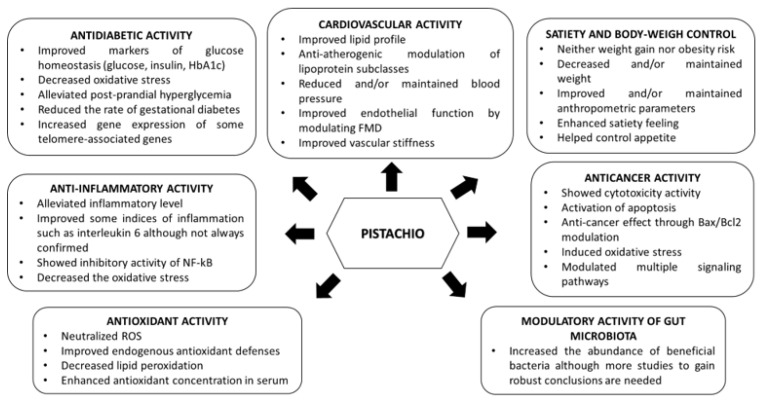
Health benefits of pistachio consumption.

**Table 1 nutrients-14-03207-t001:** Composition (g/100 g) in nutrients of dry roasted nuts. Source: USDA National Nutrient Database for Standard Reference (2020). MUFA: monousaturated fatty acids; SFA: saturated fatty acids; PUFA: polyunsaturated fatty acids.

g/100 g	Pistachio	Walnut	Almond	Hazelnut
Water	1.85	4.39	2.41	2.52
Energy (Kcal)	572	643	598	646
Lipids	45.82	60.71	52.54	62.40
SFA	5.64	5.36	4.10	4.51
PUFA	13.35	44.18	12.96	8.46
MUFA	24.53	8.37	33.08	46.61
C16:0	8.0–13.0	6.0–8.0	4.0–13.0	4.0–9.0
C18:0	0.5–2.0	1.0–3.0	2.0–10.0	1.0–4.0
C16:1	0.5–1.0	0.1–0.2	0.2–0.6	0.1–0.3
C18:1	45.0–70.0	13.0–21.0	48.0–80.0	66.0–85.0
C18:2	16.0–37.0	54.0–65.0	15.0–34.0	5.7–25.0
C18:3	0.1–0.4	13.0–14.0	N.D.	0.0–0.2
Proteins	21.05	14.29	20.96	15.03
Carbohydrates	28.28	17.86	21.01	17.60
Fiber	10.30	7.10	10.90	9.40
Sugars	7.74	3.57	4.86	4.89

**Table 2 nutrients-14-03207-t002:** Total phenolic content and antioxidant activities of different varieties of pistachios.

Type of Pistachio	Status of Pistacho	TPP	Antioxidant Assay	Value of Antioxidant Capacity	References
Larnaka (Spanish cultivar)	Natural	4900 mg/Kg fw	ORAC	330 mmol of TE/Kg	[24]
			DPPH	35 mmol of TE/Kg	
Kastel (Spanish cultivar)	Natural	1600 mg/Kg fw	ORAC	89 mmol of TE/Kg	[24]
			DPPH	13 mmol of TE/Kg	
Kerman (Spanish cultivar)	Natural	1900 mg/Kg fw	ORAC	61 mmol of TE/Kg	[24]
			DPPH	10 mmol of TE/Kg	
Uzun (Turkish cultivar)	Natural	26.2 mg/100 g fw	DPPH	8.05 μmol of TE/g	[27]
	Roasted	32.4–42.4 mg/100 g fw	DPPH	9.76–11.5 μmol of TE/g	[27]
Ohadi (Turkish cultivar)	Natural	9.23–10.55 mg/Kg GAE	DPPH	4.34–5.56 mmol TE/Kg	[28]
			ABTS	4.11–5.95 mmol TE/Kg	[28]
	Roasted	10.35–11.23 mg/Kg GAE	DPPH	3.53–6.32 mmol TE/Kg	[28]
			ABTS	5.80–7.35 mmol TE/Kg	[28]
Uzum (Turkish cultivar)	Natural	9.19–11.46 mg/Kg GAE	DPPH	7.16–13.58 mmol Trolox/Kg	[28]
			ABTS	15.69–28.28 mmol Trolox/Kg	[28]
	Roasted	10.46–12.73 mg/Kg GAE	DPPH	13.5–18.00 mmol Trolox/Kg	[28]
			ABTS	26.81–35.86 mmol Trolox/Kg	[28]

DPPH: 2,2′-diphenyl-1-picrylhydrazyl; FRAP: ferric reducing antioxidant power; ABTS: 2,2′-azino-bis(3-ethylbenzthiazoline-6-sulphonic acid); ORAC: Oxygen radical absorbance capacity; TPP: total polar phenolics; fw: fresh weight; GAE: gallic acid equivalent; TE: trolox equivalent.

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
