# Peer review of "Why Should Pistachio Be a Regular Food in Our Diet?"

_nutrients, 2022, doi:10.3390/nu14153207_

Round 1

Reviewer 1 Report

The manuscript was well prepared. Moreover, the cited and their discussion are presented in good style. However, authors have some minor points that needs to be addressed.

I feel that this manuscript is just a list of previous studies. I think there needs to be a strong message conveying the authors' original perspectives. For example, in L21~24, you indicate why nut consumption does not exceed the recommended standard, but I don't think the solution to this problem is specifically mentioned in the text.
Therefore, I feel that this is just a list of prior studies as foods that are effective but not consumed. Other than L588-593, please specify what the author believes is a strong message way to increase pistachio intake.

The arrow in Figure 1 is not properly oriented and should be re-created.

Author Response

Response to Reviewer 1 Comments

The authors would like to thank the editor and referees for their revision. We feel that their comments have greatly contributed to improving our work. The changes made in the manuscript have been marked in blue (referee 1), green (referee 2) and red (referee 3).

The manuscript was well prepared. Moreover, the cited and their discussion are presented in good style. However, authors have some minor points that needs to be addressed.

I feel that this manuscript is just a list of previous studies. I think there needs to be a strong message conveying the authors' original perspectives. For example, in L21~24, you indicate why nut consumption does not exceed the recommended standard, but I don't think the solution to this problem is specifically mentioned in the text.

Therefore, I feel that this is just a list of prior studies as foods that are effective but not consumed. Other than L588-593, please specify what the author believes is a strong message way to increase pistachio intake.

The authors would like to thank reviewer 1 for the positive comments on the quality of the manuscript.

Concerning his/her recommendation to include a clear message on how to increase pistachio intake, a series of reasoned strategies has now been commented on at the end of point 6.

The arrow in Figure 1 is not properly oriented and should be re-created.

The arrow in Figure 1 has been modified to solve the problem mentioned by the reviewer.

Reviewer 2 Report

Thank you for the opportunity to read your article. It is a well reasoned, and well researched. The majority of my comments are minor typographical/grammatical in nature. Some examples and suggestions:

Line 29: "Nuts are regarded as a relevant" as opposed to "Nuts are regard a"

Line 47: "Regarding the most consumed types of nuts" as opposed to "Regarding the type of nut more consumed"

Line 72" "However, the population does not consume" instead of "However, the population does not reach"

Again, thank you for the submission and for an insightful article.

Author Response

Response to Reviewer 2 Comments

The authors would like to thank the editor and referees for their revision. We feel that their comments have greatly contributed to improving our work. The changes made in the manuscript have been marked in blue (referee 1), green (referee 2) and red (referee 3).

Thank you for the opportunity to read your article. It is a well reasoned, and well researched. The majority of my comments are minor typographical/grammatical in nature. Some examples and suggestions:

 Line 29: "Nuts are regarded as a relevant" as opposed to "Nuts are regard a"

Line 47: "Regarding the most consumed types of nuts" as opposed to "Regarding the type of nut more consumed"

Line 72" "However, the population does not consume" instead of "However, the population does not reach"

Again, thank you for the submission and for an insightful article.

The authors would like to thank reviewer 2 for the positive comments mentioned. Both typographical and grammatical mistakes have been corrected.

Reviewer 3 Report

Tables if included, it will make the review easier to follow.

For the section 3, antioxidant phenolic composition, a brief table (just like table 1) is needed.

For the section 4, health benefits section, highly recommended to include a table.

Additionally, at the end of the manuscript, there should have an abbreviated list.

Many small mistakes:

There is no full name of CVD in the introduction.

In section 3, there are many abbreviations without full name.

Author Response

Response to Reviewer 3 Comments

The authors would like to thank the editor and referees for their revision. We feel that their comments have greatly contributed to improving our work. The changes made in the manuscript have been marked in blue (referee 1), green (referee 2) and red (referee 3).

Tables if included, it will make the review easier to follow. For the section 3, antioxidant phenolic composition, a brief table (just like table 1) is needed. For the section 4, health benefits section, highly recommended to include a table.

Thank you for your comments. As the reviewer has suggested, a new table (Table 2) has been included in section 3. However, complete information has been offered about each article commented on in section 4. Hence, the authors disagree on the need to include a third table that could be very extensive and impractical.

Additionally, at the end of the manuscript, there should have an abbreviated list.

Thank you for your appreciation. A list of abbreviations has been added at the end of this manuscript. In addition, the text has been revised to ensure that the full name is indicated the first time the abbreviation appears.

Many small mistakes: There is no full name of CVD in the introduction. In section 3, there are many abbreviations without full name.

As mentioned before, it has been possible to correct these errors after revising the entire manuscript. Thank you for your careful review.

Round 2

Reviewer 3 Report

The authors included table as suggested. But please check the whole text carefully. for example, line 178, the authors repeat the full name of CVD while they have already abbreviated in Introduction part.

Author Response

Response to Reviewer 3 Comments

The authors included table as suggested. But please check the whole text carefully. for example, line 178, the authors repeat the full name of CVD while they have already abbreviated in Introduction part.

 Again, the authors would like to thank the referee for the careful review. The manuscript has been reviewed to solve this problem.